# Quality of Systematic Reviews of the Foods with Function Claims in Japan: Comparative Before- and After-Evaluation of Verification Reports by the Consumer Affairs Agency

**DOI:** 10.3390/nu11071583

**Published:** 2019-07-12

**Authors:** Hiroharu Kamioka, Kiichiro Tsutani, Hideki Origasa, Takahiro Yoshizaki, Jun Kitayuguchi, Mikiko Shimada, Yasuyo Wada, Hiromi Takano-Ohmuro

**Affiliations:** 1Faculty of Regional Environment Science, Tokyo University of Agriculture, 1-1-1 Sakuragaoka, Setagaya-ku, Tokyo 156-8502, Japan; 2Tokyo Ariake Medical and Health Sciences University, 2-9-1 Ariake, Kouto-ku, Tokyo 135-0063, Japan; 3Division of Biostatistics and Clinical Epidemiology, University of Toyama School of Medicine, 2630 Sugiya, Toyama City, Toyama 930-0194, Japan; 4Faculty of Food and Nutritional Sciences, Toyo University, 1-1-1 Izumino, Itakura Town, Gunma 374-0193, Japan; 5Physical Education and Medicine Research Center Unnan, 328 Uji, Unnan City, Shimane 699-1105, Japan; 6Department of Nutrition, Faculty of Health Care, Kiryu University, 606-7 Asami, Midori City, Gunma 379-2329, Japan; 7Department of Food and Nutrition, Faculty of Human Life, Jumonji University, 2-1-28 Sugasawa, Niiza City, Saitama 352-8510, Japan; 8Research Institute of Pharmaceutical Sciences, Musashino University, 1-1-20 Aramachi, Nishitokyo City, Tokyo 202-8585, Japan

**Keywords:** quality, systematic review, health claim, supplement

## Abstract

**Background:** In Japan, a new type of foods with health claims, called Foods with Function Claims (FFC), was introduced in April 2015 in order to make more products available that are clearly labeled with certain health functions. Regarding substantiating product effectiveness, scientific evidence for the proposed function claims must be explained by systematic reviews (SRs), but the quality of SRs was not clear. The objectives of this review were to assess the quality of SRs based on the FFC registered on the Consumer Affairs Agency (CAA) website in Japan, and to determine whether the CAA’s verification report in 2016 was associated with improvement in the quality of SRs. **Methods:** We evaluated the reporting quality of each SR by the AMSTAR checklist on methodological quality. We searched the database from 1 April to 31 October 2015 as the before-SR and from 1 July 2017 to 31 January 2018 as the after-SR. **Results:** Among the 104 SRs reviewed, 96 final products were included: 51 (53.1%) were supplements, 42 (43.8%) were processed foods without supplements, and 3 (3.1%) were fresh foods. Of the 104 SRs, 92 (88.5%) were qualitative reviews (i.e., without meta-analysis) and 12 (11.5%) performed a meta-analysis. The average quality score of before-SRs and after-SRs was 6.2 ± 1.8 and 5.0 ± 1.9, respectively, a statistically significant decrease (*p* < 0.001). **Conclusion:** Overall, the methodology and reporting quality of after-SRs based on the FFC were poorer than those of before-SRs. In particular, there were very poor descriptions and/or implementations of study selection and data extraction, search strategy, evaluation methods for risk of bias, assessment of publication bias, and formulating conclusions based on methodological rigor and scientific quality of the included studies.

## 1. Background

The Codex Alimentarius Commission (CAC) is an intergovernmental organization that was founded in 1962 to develop food standards, guidelines, and codes of practice [1]. The basic principles of CAC are that health claims should be substantiated by currently sound and sufficient scientific evidence, provide truthful and nonmisleading information that consumers can use to choose healthy diets, and be supported by specific consumer education [2].

In accordance with CAC guidelines, only government-approved Foods for Specified Health Uses (FOSHU) and foods with nutrient function claims (FNFC) can make function claims on food labels in Japan, and these must comply with specifications and standards designated by the government [3]. The FOSHU are scientifically accepted for their usefulness in maintaining and promoting health and are therefore permitted to contain food effects and safety claims that have been evaluated by the government.

Foods approved as FNFC can be used to supplement or complement the nutrients (vitamins, minerals, etc.) that are in insufficient quantities in an individual’s daily diet. These foods can carry a nutrient function claim prescribed by the government standards and can be freely manufactured and distributed without any permission from or a notification to the national government [3]. In addition to these categories, a new type of foods with health claims, called Foods with Function Claims (FFC), was introduced in April 2015 (Figure 1). The FFC allows manufacturers to submit labeling to the Secretary-General of the Consumer Affairs Agency (CAA) in Japan that indicates the food is expected to have a specific effect on health, except for reducing the risk of diseases.

Unlike the strict evaluation criteria applied through the FOSHU and FNFC processes, the FFC is only a notification system in which food manufacturers must meet five unique and specific criteria (Table 1). Although the government does not evaluate the safety and effectiveness of the submitted product, i.e., it does not utilize a notification system, the industry (applicant) must fulfill several procedures to submit a notification. All the FFC criteria submitted by the manufacturers are disclosed on the website of the CAA, which gives approval for the labeling of food products. For a food product to claim effectiveness on its label, evidence for its proposed function claims must be substantiated by one of two standard scientific methods: clinical trials such as randomized controlled trials (RCTs), or systematic reviews (SRs). Details about the use of these two methods for food with function claims have been published on the CAA website [4]. A notable point in this system is that not only RCTs but also SRs are permitted. Since promoting deregulation is a national goal, SRs have the advantage of being easy to report to small and medium-sized enterprises because they are less expensive than RCTs.

The methodology of an SR with or without meta-analysis may not be familiar to general or nutritional researchers in the food industry. An SR addresses a question that is carefully formulated to be answered by analysis of all available evidence. It performs an objective literature search by applying predetermined inclusion and exclusion criteria to critically appraise what literature is relevant [5]. The SR is an important method that can help researchers to identify evidence of an effective intervention from a large volume of published biomedical literature.

However, although the methodology of an SR is important in terms of evidence-based nutrition (EBN), an SR has the weakness that assessment of fresh foods that most people eat daily is very difficult. Additionally, an SR may be of limited use if the methods used to conduct the SR are flawed and reporting of the SR was incomplete [6]. Moreover, the scientific validity of an SR is based on deductive planning and clear documentation of the methodological approach that was employed to design and conduct the SR [7]. We were interested in evaluating whether or not SRs of the FFC, which is based on a notification system, had been conducted by appropriate scientific methods, and we hoped to formulate a research challenge for future SRs of the FFC.

In our previous study [8], we adopted a well-known measurement tool for the ‘assessment of multiple SR’ (AMSTAR checklist) [9] and assessed the quality of 49 SRs that were based on the FFC registered on the CAA website from 1 April to 27 October 2015. Results from that study showed that the methodology and reporting quality of SRs were in the poor description category (mean ± SD: 6.2 ± 1.8 points, range 2–11 points for 11 points full-mark). Based on scientific quality, the SRs had very poor descriptions and/or implementation of the registration, poor evaluations of publication bias, and questionable conclusions.

On the other hand, the CAA in 2016 formatted the expert working group (methodologists for SR) in order to extract issues for appropriate operation of the FFC system and perform a verification [10] according to the PRISMA [11]. Fifty-one submitted SRs were selected for evaluation of quality. These SRs were all registered on the CAA website from 1 April to 31 October 2015. To complete basic standard-level SRs, considering the difficulty in handling foods, this project team attached “appropriate description in SRs based on the ‘PRISMA Checklist: an extended version for submitted SRs of Foods with Function Claims’” to its final report and included detailed proposals and examples. Most authors of the present study (HK, HO, TK, JK, MS, and HTO) also participated in this CAA project. Since the report was a specific guideline to perform and submit new SRs and was submitted to food business operators in Japan, we assumed that all researchers in this field watched it closely, followed the checklist, and completed an appropriate description afterward. Therefore, we hypothesized that the CAA’s 2016 verification report based on PRISMA [10], in addition to our article on quality evaluation in 2017 [8], were associated with improvement in the quality of subsequently submitted SRs.

The present study design was based on a previous comparative before- and after -evaluation in which RCTs published in 1994 (pre-CONSORT) were compared with RCTs published in the same journals in 1998 (post-CONSORT) [12].

The objectives of this review were to assess the quality of SRs based on the FFC registered on the CAA website in Japan, and to determine whether the CAA’s verification report in 2016 was associated with improvement in the quality of SRs.

## 2. Methods

### 2.1. Scope of This Review

The basic scientific approach of the FFC system ensures safety, functionality, and effectiveness. The purpose of this study was to assess only the quality of SRs, and it therefore focused on face and contact validity for measuring the methodological quality of SRs. Whether each product or functional substance involved is effective is a separate research issue and was not in the scope of this study.

The PRISMA statement is a respected reporting guideline designed to improve the completeness of an SR report [11]. Furthermore, although there was a new critical appraisal tool for SRs (AMSTAR 2) [13], we already performed the before-evaluation based on the AMSTAR checklist [9]. Therefore, we performed the after-evaluation with the same tool in order to compare the evaluation results.

### 2.2. Criteria for Considering Studies Included in This Review

Criteria for considering studies that were included in this review were based on those in the predefined protocol.

### 2.3. Types of Studies

Studies were eligible if they were SRs (with or without a meta-analysis).

### 2.4. Types of Participants 

This study was a review based on SRs and was therefore restricted to original SRs of healthy adults (people not suffering from any disease).

### 2.5. Comparator(s)/Control

In the original SRs, controls were defined as healthy adults identified from preplanned stratified analyses of (a) placebo controls or waiting list controls, (b) intervention groups that compared different types of products or ingredients, and (c) low- or medium-level intake groups of the same product or ingredient.

### 2.6. Types of Intervention and Language

For processed foods in the form of supplements, studies included at least one intervention group in which the functional ingredient and the final product were applied. For fresh food and other processed food, studies included at least one exposure group in which the functional ingredient and the final product were applied and included observational studies and intervention studies.

### 2.7. Types of Outcome Measures 

Outcome measures included many types of positive contributions to health, to the improvement of a function, or to preserving health as an outcome. In effect, we included all notified SRs.

### 2.8. Search Methodology for Identification of Studies

#### 2.8.1. Search Strategies

Our search of the databases on the CAA website covered the period from 1 April 2015 (starting date) through 27 October 2015 for the before-SRs [8], and from 1 July 2017 to 31 January 2018 for the after-SRs. The special search strategies contained the elements and terms (i.e., a specific search method based on keywords) on the CAA website. All references in identified SRs were screened. The search was performed by the steering author (HK).

#### 2.8.2. Hand-Searching and Reference Checking

Since this study was limited to SRs registered in the CAA database, hand-searching and reference checking were not applicable.

### 2.9. Review Methods

#### 2.9.1. Selection of Studies

To select the studies that were to be reviewed, all criteria were applied by the steering author (HK) to the full library of articles published on the CAA website. Studies were selected when (i) the design was an SR based on an intervention study, (ii) the study was appropriately notified by the CAA, and (iii) the study was published on the website. Studies (notification) that were excluded are presented with reasons for exclusion.

#### 2.9.2. Quality Assessment of Included Studies

To ensure that variation was not caused by systematic errors in study design or execution, four review authors (YW, TY, MS, and JK) independently assessed the quality of articles (i.e., every two reviewers were paired). Disagreements and uncertainties were resolved by discussion with another author (HK). A full quality appraisal of these papers was made using a combined tool based on the AMSTAR checklist that was developed to assess methodological quality of SRs. Each item was scored as ‘present’ (Yes), ‘absent’ (No), ‘unclear or inadequately described’ (Can’t answer), or ‘Not applicable’ (n/a). Depending on the study design (with or without meta-analysis), some items were not applicable; therefore, the “n/a” score was not considered to be an error in the calculation for quality assessment. Although the original AMSTAR has 11 check items, two meanings can be applied to item #3, which reads as follows: “Was a comprehensive literature search performed. At least two electronic sources should be searched. The report must include years and databases used. Key word and/or MESH terms must be stated and where feasible the search strategy should be provided (continued).” Because the guideline for FFC notification on the CAA website requires the use of at least two electronic databases, we divided #3 into two parts in order to detect any trend arising from the use of databases: “#3a; which databases did the SR use or number of the other databases”, and “#3b; Did the SR use MESH terms and related search function to detect comprehensively”.

All authors attended one 3-hour consensus-training session based on the AMSTAR checklist before starting the quality assessment to ensure that they used the same criteria and correctly evaluated the check-items for an SR.

The percentage of descriptions present on all 11 (excluding #3a) of the check items for the quality assessment of articles was determined. Then, based on the percentage of risk of poor methodology and/or bias, each item was assigned to one of the following categories: good description (80%–100%), poor description (50–79%), or very poor description (0–49%).

Disagreements and uncertainties were resolved by discussion with other authors (HK, TK, and HO). Interrater reliability was calculated by the steering author (HK) on a dichotomous scale using percentage agreement and Cohen’s kappa coefficient (*k*).

#### 2.9.3. Characteristics of Studies and Data Extraction

Two authors (HK and HO) described the characteristics from each article based on information on the CAA website but did not produce a structured abstract for SRs, which is recommended [14]. Because this study focused on evaluating quality of SRs of the FFC, it did not summarize evidence for the effectiveness of each SR.

#### 2.9.4. Research Protocol Registration

We submitted and registered our research protocol to PROSPERO (CRD42017080833) and UMIN-CTR (UMIN000029821). PROSPERO is an international database of prospectively registered SRs in health and social care [15]. Key features from a review protocol are recorded and maintained as a permanent record in PROSPERO. UMIN-CTR is a Japanese and international database of prospectively registered clinical trials and other trials with SRs in health and social care and was accepted as an international registry database by the International Committee of Medical Journal Editors in 2007. In a previous study [8], we implemented our protocol before UMIN-CTR was formally launched on 1 April 2015, and we planned to continue checking target SRs prospectively from our study start date. In the present study, we also planned to continue reviewing target SRs prospectively from 1 July 2017 to 31 January 2018.

#### 2.9.5. Statistical Analysis

A two-sample *t* test was employed for comparisons between two terms (number of databases and before- and after-evaluation scores) with continuous variables in the analysis. The χ^2^ test and Fisher’s exact test were performed with discrete variables (i.e., number and % of good description on each item). Statistical analysis was performed with SPSS version 23.0 (IBM Corporation, Armonk, NY, USA) for Windows. For all analyses, *p*-values less than 0.05 were considered statistically significant.

## 3. Results

### 3.1. Study Selection and Characteristics 

Of the 294 potentially relevant articles included in the literature search, 198 notifications were excluded because they did not meet the eligibility criteria (Figure 2). A total of 104 SRs (including eight multiple claims) met all inclusion criteria. The language of all eligible publications was Japanese.

The contents of all articles were summarized as brief characteristics (Table 2). Among the 104 SRs reviewed, 96 final products were included: 51 (53.1%) were supplements, 42 (43.8%) were processed foods without supplements, and 3 (3.1%) were fresh foods. Of the 104 SRs, 91 (87.5%) were qualitative reviews (i.e., without meta-analysis), and 13 (12.5%) performed a meta-analysis.

### 3.2. Quality Assessment

We evaluated 11 items from the AMSTAR checklist in more detail (Table 3 and Appendix A). Interrater reliability metrics for the quality assessment indicated substantial agreement (71.7%, *k* = 0.558) for all 1144 items (11 items multiplied by 104 SRs).

Overall, there was an increase over time in evaluation score. The average of the quality score for before-SRs and after-SRs was 6.2 ± 1.8 and 5.0 ± 1.9, respectively, which was a statistically significant decrease (*p* < 0.001).

Regarding category of each item in the after-evaluation group, four items (#1, #5, #6, and #9) were a good description (80%–100%), one item (#7) was a poor description (50%–79%), and six items (#2, #3b, #4, #8, #10, and #11) were a very poor description (0%–49%).

There was a good description and/or implementation for the following items: “Was an ‘a priori’ design provided?” (before-, 4% and after-, 86%, *p* < 0.001); “A list of included and excluded studies should be provided.” (before-, 100% and after-, 98%, *p* = 0.329); “Were the methods used to combine the findings of studies appropriate?” (before-, 56% and after, 92%, *p* = 0.116); and “Were the characteristics of the included studies provided?” (before-, 84% and after-, 88%, *p* = 0.521). These items were still a good description or improving in the after-evaluation.

There continued to be a poor description and/or implementation for the item, “Was the scientific quality of the included studies assessed and documented?” (before-, 73% and after-, 59%, *p* = 0.076).

The other items were a very poor description and/or implementation: “Was there duplicate study selection and data extraction?” (decreased from 65% to 41%, *p* < 0.01); “Did the SR use the MESH terms and related search function to detect comprehensively?” (decreased from 53% to 47%, *p* = 0.492); “Was the status of publication used as an inclusion criterion?” (decreased from 24% to 3%, *p* < 0.001); “Was the scientific quality of the included studies used appropriately in formulating conclusions?” (decreased from 27% to 26%, *p* = 0.94); “Was the likelihood of publication bias assessed?” (increased from 12% to 13%, *p* = 0.964); and “Was the COI stated?” (all decreased; from 78% to 25%, *p* < 0.001).

According to one component of #3a, “which databases did the SR use or number of the other databases?”, the number of used databases was the same between before- and after-evaluation (mean ± SD was 3.8 ± 1.8, *p* = 1.000). According to respective before- and after-evaluations, the high utility databases were PubMed (93.9% and 100%), JDream III (in Japanese databases, 79.6% and 60.6%), Ichushi-Web (in Japanese databases, 67.3% and 64.4%), The Cochrane Library (with CENTRAL, 49.0% and 55.8%), and UMIN-CTR (Japanese clinical trial registry, 18.4% and 23.1%).

## 4. Discussion

This is the first prospective before- and after- SR of SRs of the FFC registered on the CAA website in Japan.

The FFC in Japan is an original and unique system regarding health claims. A food business operator must submit a completed notification and related documents to the Secretary-General of the CAA 60 days prior to the launch date. Therefore, all consumers can check all content such as safety, functional mechanism, and effectiveness (i.e., total evidence) of the product, resulting in high transparency.

We propose that this study will be helpful to researchers and government officials who want to know about new health claims in advanced countries. We expected that the total quality of after-SRs might be improved significantly by the CAA’s verification report in 2016 [10], but this study instead showed deterioration in quality. Therefore, it is necessary to discuss the interpretation of these findings and propose a practical future strategy for this issue.

### 4.1. Quality Assessment of Target SRs

Overall, the quality of articles significantly decreased in conduct and reporting. Although four items (#1, #5, #6, and #9) in the after-SRs group were improving or remained a good description, another seven items were poor or very poor.

The methodology for most SRs did not attempt to include so-called grey literature by the use of many other types of databases and classical literature searches. Grey literature was defined here as studies that are unpublished, have limited distribution, and/or are not included in the bibliographical retrieval system [16]. The importance of including grey literature in all SRs has been previously discussed [17]. Implementers of SRs need to recognize the importance of also searching grey literature.

“Was the likelihood of publication bias assessed?” also remained flawed in the assessment process. Additionally, many SRs used “at least two electronic sources”, but these were only Japanese databases and/or not the more traditional English databases like EMBASE or MEDLINE. Furthermore, it has been pointed out that there is a bias in coverage with only one database (i.e., PubMed) [18,19]. Researchers performing SRs therefore need to use multiple databases.

Publication bias remains an area of contention amongst researchers who assess the quality of SRs [20,21]. However, it remains a research priority because it is unclear what impact publication bias has on making decisions in healthcare [9]. We assume that the new FFC guideline provides a better description of how to assess publication bias, especially for a qualitative SR.

“Was the conflict of interest stated?” was a serious problem. Although most SRs described a part of the COI, they did not include all necessary information such as the SR’s sponsor, SR’s funding, author’s affiliation, SR’s outsourcing information (research agency), supervision allowance, and consulting fees for an SR. In fact, the targeted SRs included those that were conducted only by the company itself, those conducted by other companies, such as raw material makers, those conducted by a research agency, and those supervised by academia researchers. We assume that the primary reason reviewers of quality assessment judged many SRs as ‘unclear or inadequately described’ (Can’t answer) was because they could not cover these elements properly. The International Committee of Medical Journal Editors (ICMJE) emphasizes that when authors submit a manuscript of any type or format, they are responsible for disclosing all financial and personal relationships that might bias or be seen to bias their work [22].

“Was there duplicate study selection and data extraction?” was also a very poor description. This was described clearly in the CAA’s verification report in 2016 [10]. Because everyone makes mistakes occasionally, there should be at least two independent data extractors, and a consensus procedure for disagreements should be in place. It was not clear why two additional researchers did not perform independent assessments for some of the SRs.

“Did the SR use the MESH terms and related search function to detect comprehensively?” got worse in the after-evaluation. The guideline [4] instructs that “To search comprehensively, a search formula made by combining free items and controlled terms (including MeSH for PubMed) appropriately will be set per bibliographic databases.” In addition, the report [10] points out that it is essential to design an optional search formula by combining keywords and thesauruses (such as MeSH) appropriately for each clinical question according to each database characteristic.

We assume that there were multiple reasons for the quality declining significantly. The FFC system is just a notification, so the CAA does not evaluate the safety and effectiveness of a submitted product. The number of notifications has increased since the system was launched in 2015 to lead sales promotions, but the reason for this might be that multiple companies had purchased copies of the SRs that had already been accepted by the CAA and submitted them to the CAA as basis material for the evidence. Therefore, many low-level SRs may have been contained in the FFC system, so the quality of after-evaluation might have deteriorated.

### 4.2. Validity and Reliability of Quality Assessment by AMSTAR Checklist

For the before-evaluation, we adopted a measurement tool used for the ‘assessment of multiple SR’ (AMSTAR checklist). The “R-AMSTAR” [23] was also developed as an approach to minimize bias of any kind in SRs. In terms of interrater reliability and validity, the AMSTAR score was very high compared with scores from other tools. In terms of feasibility, it was very appropriate that scoring time of the AMSTAR was short (between 10 and 20 min) [24]. Furthermore, a recent methodological study showed that AMSTAR and the risk of bias in systematic reviews (ROBIS) had similar interrater reliability but differed in their construct and applicability [25].

Although we had one consensus-training session and all reviewers had conducted a quality assessment of SRs more than once, the interrater reliability metrics for the quality assessment indicated substantial agreement was average 71.7%, *k* = 0.558. It can also be interpreted that there were many SRs for which the quality reviewers were confused as to whether it was a “Yes”, “No”, or “Can’t answer”.

Additionally, the reviewers seemed to have some ambiguity about the details of each item. Recently, a quality assessment check list, AMSTAR 2, was developed that allows for individual responses that do not impart judgment for each item [13]. AMSTAR 2 retains 10 of the original domains, has 16 items in total (compared with 11 in the original), has simpler response categories than the original AMSTAR, includes a more comprehensive user guide, and includes the identification of high-quality SRs. It might be useful to evaluate the detail quality for each item of the SRs in a future study.

### 4.3. Future Research Challenge to Improve the Quality of SRs of the FFC

Table 4 shows the future research challenge for studies on the health enhancement effects of the FFC and related healthy foods. We assumed that there are three important dimensions and six tasks due to improved systematic reviews. Regarding the food industry, researchers must study the current standard rule of an SR (i.e., AMSTAR 2, PRISMA checklist, and PRISMA-NMA checklist for meta-analysis) [26] before research is conducted. If an applicant is concerned about the implementation of an SR, they should immediately consult with experts on research methodology, which will avoid creating inappropriate SRs. Moreover, since the CAA only performs formal confirmation of documents, the methodology of the SRs that had already been notified was not always correct. Therefore, if another company’s SR is reused for a notification, it becomes necessary for an applicant to carefully examine the SR before deciding to confidently introduce its own product to the market.

Academia should provide its own support for food companies and other companies to implement the SR properly, and academic researchers will need to continue to convey appropriate SR methodologies to the food industry. In the present study, it became clear that there were many methodological deficiencies in targeted SRs. The FFC system in Japan relies on one SR or one clinical trial, such as a RCT as a basis for efficacy. However, a Japanese research group recently identified problems with the reporting quality and associated issues for RCTs of the FFC [27]. There was insufficient information on items associated with sample size, allocation and blinding, results of outcomes and estimation, generalizability of the results, and study registration numbers. Because it is a notification system, it is essential for academic researchers, including our group, to monitor all SRs and clinical trials for the FFC.

Considering that the Japanese government has introduced the world’s most advanced FFC system as part of its growth strategy (i.e., deregulation), it may be difficult for the CAA to review individual SRs. Therefore, to protect consumers, we assume it is necessary to confirm that the notification SR is above a certain level of quality.

Either way, even for an SR that has already been accepted, it will be necessary to issue the latest (updated) version 5 to 10 years later. The prospect of this future requirement will encourage all existing SRs to be conducted by scientifically correct methodologies.

## 5. Limitations

This review had several limitations that should be acknowledged. First, publication bias was possible because there was not enough use of multiple databases for each SR. Second, we could not perform an evaluation using the PRISMA checklist. Third, our study design focused on the quality of SRs; therefore, we could not validly assess the safety or the functional mechanism of any of the products reviewed in the SRs. Lastly, because we did not conduct a retrospective analysis of the quality of “primary studies cited or used as references” that were described in submitted SRs, the effectiveness of functional substances or finished products could not be addressed.

## 6. Conclusions

Overall, the quality of methodology and reporting in after-SRs based on the FFC was poorer than that based on before-SRs. In particular, there were very poor descriptions and/or implementation of study selection, data extraction, search strategy, evaluation methodology for risk of bias, assessment of publication bias, and formulating conclusions based on methodological rigor and scientific quality of the included studies.

To develop SRs of the FFC and launch a similar global food claim notification system, the following factors will be important: (i) applicants will need to use some global standard checklist such as AMSTAR 2, PRISMA, or PRISMA-NMA; (ii) applicants will need to critically examine the quality when using another applicant’s SR; (iii) academic researchers should support the food industry in order to perform an SR and/or clinical trial properly; and (iv) country authorities should confirm that the notification SR is above a certain level of quality.

## Figures and Tables

**Figure 1 nutrients-11-01583-f001:**
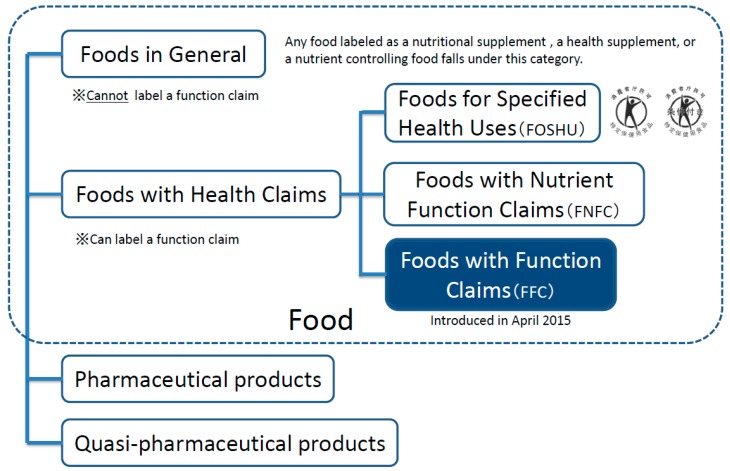
Food labeled with certain nutritional or health functions in Japan.

**Figure 2 nutrients-11-01583-f002:**
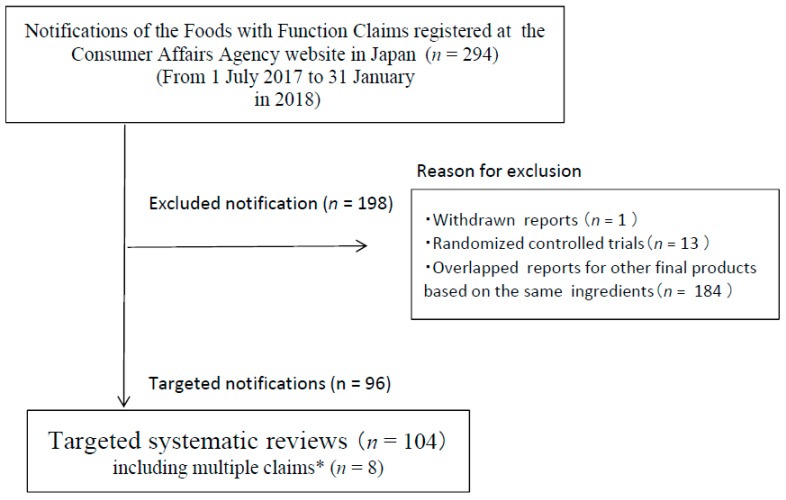
Flowchart of trial process. * A product has more than one functionality (e.g., decreased body fat and increased HDL cholesterol) and is displaying them.

**Table 1 nutrients-11-01583-t001:** Characteristics of the foods with function claims in Japan.

1	Foods with Function Claims are for people not suffering from any disease (excluding minors, pregnant women (and those planning a pregnancy), or lactating women).
2	All food products including fresh produce are subject to this system.*
3	Prior to market entry (before at least 60 days), food business operators are required to submit information, such as on food safety and effectiveness and the system in place to collect information on adverse health effects, to the Secretary-General of Consumer Affairs Agency.
4	Unlike Foods for Specified Health Uses, the government does not evaluate the safety and effectiveness of the submitted product, i.e. notification system.
5	The submitted all information is disclosed on the website of the Consumer Affairs Agency.**

Modified partially for this study based on the Consumer Affairs Agency website in Japan. * Excluding Foods for Special Dietary Uses (including FOSHU), FNFC, alcohol-containing beverages, and food products that may lead to the excessive consumption of fat, cholesterol, sugar (limited to mono- and disaccharides, excluding sugar alcohols), or sodium. ** The all information was only written in Japanese.

**Table 2 nutrients-11-01583-t002:** Characteristics of systematic reviews of the foods with function claims.

No. *	Product Name	Food Business Operator	Classification of Food 1 Supplement 2 Processed Food 3 Fresh Produce	Functional Substance
C48	Ayumu Chikara (Fruit Yogurt)	Lion Corporation	2	HMB(β-hydroxy-β-methylbutyrate), Glucosamine hydrochloride
C49	Kangengata CoQ10 150	FINE JAPAN Co., Ltd.	1	Reduced coenzyme Q10
C50	DESK RAKU	Nihon kefir Co., Ltd.	1	Lutein
C51	OFURO TIME cocktail taste salty dog taste	KING BREWING Co., Ltd.	2	Indigestible dextrin (dietary fiber)
C52	OFURO TIME cocktail taste cassis orange taste	KING BREWING Co., Ltd.	2	Monoglucosyl hesperidin
C53	Karada ni yasashii mizu (lemon taste)	Melodian Co., Ltd.	2	Hyaluronic acid Na
C54	Karada ni yasashii mizu (peach taste)	Melodian Co., Ltd.	2	Indigestible dextrin (dietary fiber)
C55	Karada ni yasashii mizu (grapefruit taste)	Melodian Co., Ltd.	2	Salacinol made from Salacia
C59	The product has not been named in English yet.	Kyoto Pharmaceutical Industries, Ltd.	1	G biloba flavonoid glycoside, G biloba terpene latone
C60	DHA no kiwami 1000 mg plus	Bizen Chemical Co., Ltd.	1	DHA, EPA
C61	SUNKINOU Fish oil	Sunsho Pharmaceutical Co., Ltd.	1	DHA, EPA
C62	Kotsukotsu Soybean Isoflavone	KYOWA YAKUHIN Co., Ltd.	1	Soy isoflavone
C63	DHA no kiwami 1000 mg plus W	Bizen Chemical Co., Ltd.	1	DHA, EPA
C65	EGAO GABA Stress Care	EGAO Co., Ltd.	1	GABA (γ-Aminobutyric acid)
C68	SUNKINOU Bifidobacteria	Sunsho Pharmaceutical Co., Ltd.	1	Bifidobacterium longum BB536
C69	Yumemin	SIMANOYA Co., Ltd.	1	L-Theanine
C70	meiji GABA COFFEE	Meiji Co., Ltd.	2	GABA
C71	Glucosamine2000 KAIHO	NIHON_YAKUSHIDO Co., Ltd.	1	Glucosamine hydrochloride
C72	Ginkgo Extract	Yuuki Medicine manufacture, Inc.	1	G biloba flavonoid glycoside, G biloba terpene latone
C73	Soup with Brown rice Chinese soup	SHOKKYO Co., Ltd.	2	Indigestible dextrin (dietary fiber)
C75	Ginkgo leaf	Morishita Jintan Co., Ltd.	1	G biloba flavonoid glycoside, G biloba terpene latone
C78	Lutein	KYOWA HAKKO BIO Co., Ltd.	1	Lutein
C80	Peptide Meinte	FUJI OIL Co., Ltd.	2	Seryl tyrosine made from soybean
C82	The product has not been named in English yet.	Mikakuto Co., Ltd.	2	GABA
C84	KAGOME tomato juice “kou-lycopene tomato shiyou, syokuen-mutenka”	KAGOME Co., Ltd.	2	Lycopene, GABA
C85	Chanson Bilberry Plus	Chanson Cosmetics inc.	1	Bilberry extract anthocyanin
C86	MAINICHI KORE 1HON EPA+DHA Fish Sausage 50	Nippon Suisan Kaisha, Ltd.	2	DHA, EPA
C90	Health Fit Cha	Ginza stefany Inc.	2	Indigestible dextrin (dietary fiber), Isoflavone made from kudzu (Tectorigenin)
C91	SAGERU	Creare Co., Ltd.	1	GABA
C95	The product has not been named in English yet.	Mikakuto Co., Ltd.	2	Acetic acid
C96	KOTHARAEX TSUBU	Fuji Sangyo Co., Ltd.	1	Neokotalanol
C97	Gussumin GABA no Chikara	Lion Corporation	2	GABA
C99	Premier Rich Perfect Asta Hyaluronic acid Powder	Asahi Group Foods, Ltd.	1	Hyaluronic acid Na
C103	SUNKINOU Ceramide	Sunsho Pharmaceutical Co., Ltd.	1	Glucosylceramide made from pineapple
C105	EYE GUARD	NATURALLY HEALTH FOODS Co., Ltd.	1	Lutein, Zeaxanthin
C109	Organic Enseki Kale	Enseki Aojiru Co., Ltd.	2	GABA (γ-Aminobutyric acid)
C110	Pep-Gyu	YOSHINOYA Co., Ltd.	2	Valine-Valine-Tyrosine-Proline made from globin
C111	Daizu peptide Genen shoyu (dashi-iri)	Kikkoman Food Products Company	2	Soybean peptide
C112	Clear Lutein	Yazuya Co., Ltd.	1	Lutein
C114	Latwell	ASAHI CALPIS WELLNESS Co., Ltd.	1	Lactotripeptides (Valine-Proline-Proline/Isoleucine-Proline-Proline)
C120	Suupu you ito kanten	Ina Food Industry Co., Ltd.	2	Galactan made from agar (fiber)
C121	Ginkgo biloba leaf Extract Tablet	AFC Co., Ltd.	1	G biloba flavonoid glycoside, G biloba terpene latone
C123	Hiaruronsan C-jelly	Earth Corporation	2	Hyaluronic acid Na
C124	Beau Clair	HEALTH RESEARCH FOUNDATION	1	Lutein, Zeaxanthin
C125	Vegetable lactobacillus TAKUMINO-KIMCHI	Tokai Pickling Co., Ltd.	2	L. plantarum TK61406
C126	HAKKOU TSUBAKICHA containing indigestible dextrin	YAMACHIYA Co., Ltd.	2	Indigestible dextrin (dietary fiber)
C127	Kiokuryokusengen	Aishitoto.Co., Ltd	1	G biloba flavonoid glycoside, G biloba terpene latone
C130	Routeri yogurt	OHAYO DAIRY PRODUCTS Co., Ltd.	2	L.reuteri DSM 17938
C134	Bifistock	EVERLIFE Co., Ltd.	1	Bifidobacterium lactis HN019
C135	ICHOBA	MORIKAWA KENKODO Co., Ltd.	1	G biloba flavonoid glycoside, G biloba terpene latone
C138	arukumikata	Halmek Corporation	1	Undenatured type II collagen
C161	HMB100	Faveurmarche Co., Ltd.	1	3-Hydroxy 3-MethylButyrate (HMB)
C162	Everyday Dairy yogurt <Low-Fat>	Nippon Dairy Co-operated Co., Ltd.	2	Lactobacillus bifidus BB-12 (B. lactis)
C166	Healthy Plus Sarasara Mugicha	ITO EN, LTD.	2	Monoglucosyl hesperidin
C176	Suyasuya rerakkusui	Medione	1	Hyperoside made from Lafua, Isoqueritrin made from Lafuma
C179	Hitomi Management	QOL Laboratories, Inc.	1	Bilberry extract anthocyanin
C180	Lutein Hitomi no Kagayaki	KOHKAN Pharmaceutical Institute Co., Ltd.	1	Lutein
C181	Ichimokuryozen W	SOCIA Co., Ltd.	1	Lutein
C182	aojiruzanmai akatsuki	TV SHOPPING LABORATORY Co., Ltd.	1	L-Theanine
C183	The product has not been named in English yet.	Mizkan Co., Ltd.	2	Acetic acid
C192	Nyusankin shokora	LOTTE Co., Ltd.	2	Lactobacillus brevis NTT001
C197	Hiroshima Mikan	Hiroshima Pref. Fruit Growers Cooperative Association	3	β-Cryptoxanthins
C200	Body Challenge	Ryusendo Co., Ltd.	1	Ellagic acid made from African mango eaves
C201	MUSENMAI GABA KOSHIHIKARI	ZEN-NOH PEARL RICE Co., Ltd.	2	γ-Aminobutyric acid (GABA)
C205	Suntory RURU-CHA	Suntory Beverage & Food Limited	2	Inulin
C207	Ginkgo Biloba EX	Amway Japan G.K.	1	G biloba flavonoid glycoside, G biloba terpene latone
C208	Mental Balance Chocolate GABA <Bitter>Mobile Type	EZAKI GLICO Co., Ltd.	2	γ-Aminobutyric acid
C209	Melax eye	Yawata Corporation	1	Lutein
C213	Aminomin N	Pharma Foods International Co., Ltd.	1	GABA
C216	Kiokuru	Suppleplus Family Co., Ltd.	1	G biloba flavonoid glycoside, G biloba terpene latone
C218	EasyTablet TERMINALIA	EC STUDIO Co., Ltd.	1	Gallic acid made from Terminal nari abe lyrica
C222	Supplement Joiner	Shiseido Company, Limited	1	Salmon nasal proteoglyan
C228	Moisture + Honey-bush Blend Tea	SHOWA PHARMACEUTICAL Co., Ltd.	2	N-Acetylglucosamine
C229	megumi Gasseri SP Strain Yogurt Drink-type Berry Mix 100g	MEGMILK SNOW BRAND Co., Ltd.	2	Lactobacillus gasseri SBT2055
C230	Megami	Sun Chlorella Corp.	1	Astaxanthin
C233	Medikara supplement	Asahi Group Foods, Ltd.	1	Lutein
C242	Kenkou Benifuuki Cha	Yawata Corporation	2	Methylated catechin (epigallocatechin-3-O-[3-O-methyl] gallate)
C249	shinn oishiimushimame mushisaradamame	Maruyanagi Foods Inc.	2	Soy isoflavone
C251	Glucosamine 2000	DHC Corporation	1	Glucosamine hydrochloride
C264	Kuensan Powder	FINE JAPAN Co., Ltd.	2	Citric acid
C271	gaba megumi rice (Special Three percent milled-rice)	Tokyo foods create Co., Ltd.	3	GABA (γ-Aminobutyric acid)
C272	Ketsuatsu ga takame no kata no hakkou kuro uuron cha	FINE JAPAN Co., Ltd.	2	GABA
C273	CeramiDo?	Facelabo Co., Ltd.	1	Glucosylceramide made from rice
C275	Q’SAI lilac-01 lactic acid bacterium	Q’SAI Co., Ltd.	1	Bacillus coagulans lilac-01
C276	Todoku Tsuyosa no Nyusankin 100(Foods with function claims)	ASAHI SOFT DRINKS Co., Ltd.	2	L. gasseri CP2305
C282	Algae DHA Capsule	Nikken Sohonsha Corporation	1	DHA made from aurantiochytrium
C289	Omega A.D.E.	Suntory Wellness Limited	1	DHA, EPA, ARA
C295	Wasurerumonka	imunos Co., Ltd.	1	Phosphatidylserine made from soybean
C303	The product has not been named in English yet.	Takayuki Nishie	2	Acetic acid
C318	Mattanthermo	DHC Corporation	1	Monoglucosyl hesperidin
C321	Shinjyumai	Kometo Sangyo Kaisha, Ltd.	2	GABA
C322	soy isoflavone kodaizumoyashi	Meisui Bijin Factory Co., Ltd.	3	Soy isoflavone
C331	The product has not been named in English yet.	Fuji Chemical Industries Co., Ltd.	2	Astaxanthin
C332	webber naturals Lutein Plus	factorsgroup Japan LLC	1	Lutein
C334	HESPERIDIN & COLLAGEN	EZAKI GLICO Co., Ltd.	2	Monoglucosyl hesperidin, Low molecular collagen peptide made from fish
C339	GOMATOUNYUJITATE MINNANOMIKATA DHA	Nippon Suisan Kaisha, Ltd.	2	EPA, DHA

* This number is an identification number on the CAA website.

**Table 3 nutrients-11-01583-t003:** Quality assessment of systematic reviews (SR) of the foods with function claims.

No.	Items	Before-Verification		After-Verification		*p*-Value
		N = 49		N = 104		
#1	Was an ‘a priori’ design provided?	2	4%	89	86%	<0.001
#2	Was there duplicate study selection and data extraction?	32	65%	43	41%	0.006
#3a	Was a comprehensive literature search performed?	3.8 ± 1.8 *	(2–15)	3.8 ± 1.8 *	(2–17)	1.000
#3b	Did the SR use the MESH terms and related search function to detect comprehensively?	26	53%	49	47%	0.492
#4	Was the status of publication used as an inclusion criterion?	12	24%	3	3%	<0.001
#5	A list of included and excluded studies should be provided.	49	100%	102	98%	0.329
#6	Were the characteristics of the included studies provided?	41	84%	91	88%	0.521
#7	Was the scientific quality of the included studies assessed and documented?	36	73%	61	59%	0.076
#8	Was the scientific quality of the included studies used appropriately in formulating conclusions?	13	27%	27	26%	0.940
#9	Were the methods used to combine the findings of studies appropriate?	5/9**	56%	12/13**	92%	0.116
#10	Was the likelihood of publication bias assessed?	6	12%	13	13%	0.964
#11	Was the conflict of interest stated?	38	78%	26	25%	<0.001
Evaluation score	pts./11	6.2 ± 1.8	(2–11)	5.0 ± 1.9	(1–11)	<0.001

No (%) of good description for #1, #2, #3b, #4, #5, #6, #7, #8, #9, #10, and #11. Mean±SD (range) for #3a and evaluation score. χ^2^ test for each item, but Fisher’s exact test for #9. * Number of electronic databases. ** Article number with meta-analysis is 9 in before-evaluation and 13 in after-evaluation.

**Table 4 nutrients-11-01583-t004:** Research challenge on systematic review of the foods with function claims.

**For food industry**	
#1	The applicants should conduct research based on AMSTAR 2 checklist.
#2	The applicants should conduct research based on PRISMA checklist and PRISMA-NA (for meta-analysis).
#3	The applicants should examine its quality when using the SR of another company that had already been accepted by the CAA.
#4	The applicants should consult with academia researchers for unclear points in methodology.
**For academia**	
#5	Academic researchers should provide support for food companies and others to implement the SR properly.
□	Study plan (study selection and data extraction, search strategy, and evaluation method of bias risk)
□	Implementation (assessment of publication bias, and formulating conclusion based on methodological rigor quality)
□	Description (conflict of interest)
**For the Consumer Affairs Agency in Japan**	
#6	The authorities should evaluate not only the formal confirmation* in the document but also the quality (certain level or higher) of the SR.

* Currently, the government intends to deregulate in food industy, so the CAA cannot examine the quality of each SR.

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
