# Peer review of "Quality of Systematic Reviews of the Foods with Function Claims in Japan: Comparative Before- and After-Evaluation of Verification Reports by the Consumer Affairs Agency"

_nutrients, 2019, doi:10.3390/nu11071583_

Round 1

Reviewer 1 Report

see manuscript with comments

Author Response

Thank you for your important and insightful suggestions. We carefully considered your suggestions and revised accordingly. Moreover, an experienced editor carefully reviewed the attached paper. We appreciate the opportunity to further improve the manuscript.

Reviewer 2 Report

The aim of the work is interesting, but the quality of presentation should be improved. The following major revisions are needed.

Proper keywords should be used.

In introduction , lines 63-69 shold be enlarged.

Lines 69-71 shold be better explained.

Lines 72-75: This concept should be enlarged and references added.

The limits and advantages of SR methodology in nutrition field should be better described. 

In Material and Methods, 

In paragraph "criteria for..." some introductive lines should be inserted before subparagraphs 

Lines117-120 should be better described.

Lines 132-135 the authors should better justify the choice of controls.

The subparagraph "Search Strategies " should be improved .

The sentence in line 51 should be justified.

The description of results should be carried out with major details.

In the Discussion of results more previous works should be cited to discuss results.

Table 1 should be reorganized.

The references should be formatted following MDPI format.

Author Response

Response to Reviewer 2 Comments

Thank you for your important and insightful suggestions. We carefully considered your suggestions and revised accordingly. Moreover, an experienced editor carefully reviewed the attached paper. We appreciate the opportunity to further improve the manuscript.

Point 1: Proper keywords should be used.

Response 1: We deleted the words “Foods with Function Claims”.

Point 2: In introduction, lines 63-69 should be enlarged.

Response 2: We added the following sentence as additional explanation: “The FFC allows manufacturers to submit labeling to the Secretary-General of the Consumer Affairs Agency (CAA) in Japan that indicates the food is expected to have a specific effect on health, except for reducing the risk of diseases.”

Point 3: Lines 69-71 should be better explained.

Response 3: We added the following sentences as additional explanation: “Although the government does not evaluate the safety and effectiveness of the submitted product, i.e., it does not utilize a notification system, the industry (applicant) must fulfill several procedures to submit a notification.”

Point 4: Lines 72-75: This concept should be enlarged and references added.

Response 4: We added the following sentences as an additional explanation; however, the citation remains the same, as only the guideline is relevant reference [4]: “A notable point in this system is that not only RCTs but also SRs are permitted. Since promoting deregulation is a national goal, SRs have the advantage of being easy to report to small and medium-sized enterprises because they are less expensive than RCTs.”

Point 5: The limits and advantages of SR methodology in nutrition field should be better described.

Response 5: Thank you for your essential suggestion. According to your instruction, we added the following sentence to the Background section: “However, although the methodology of a SR is important in terms of evidence-based nutrition (EBN), a SR has the weakness that assessment of fresh foods that most people eat daily is very difficult.” 

Point 6: In Material and Methods,

In paragraph "criteria for..." some introductive lines should be inserted before subparagraphs

Response 6: Thank you for your important suggestion. We added the following sentence before the subparagraphs: “Criteria for considering studies that were included in this review were based on those in the predefined protocol.”

Point 7: Lines117-120 should be better described.

Response 7: We changed these sentences to the following: “The purpose of this study was to assess only the quality of SRs, and it therefore focused on face and contact validity for measuring the methodological quality of SRs. Whether each product or functional substance involved is effective is a separate research issue, and was not in the scope of this study. ”

Point 8: Lines 132-135 the authors should better justify the choice of controls.

Response 8: We added and/or revised the following sentences: “This study was a review based on SRs, and was therefore restricted to original SRs of healthy adults. In the original SRs, controls were defined as healthy adults identified from pre-planned stratified analyses of a) placebo controls or waiting list controls, b) intervention groups that compared different types of products or ingredients, and c) low- or medium-level intake groups of the same product or ingredient.”

Point 9: The subparagraph "Search Strategies" should be improved.

Response 9: We added and revised the following sentences: “Our search of the databases on the CAA website covered the period from 1 April 2015 (starting date) through 27 October 2015 for the before-SRs [8], and from 01 July 2017 to 31 January 2018 for the after-SRs.”

Point 10: The sentence in line 151 should be justified.

Response 10: We revised the following sentences: “Since this study was limited to SRs registered in the CAA database, hand-searching and reference checking were not applicable.”

We also deleted the following sentence: “We did not need to perform any handsearches.”

Point 11: The description of results should be carried out with major details.

Response 11: Since this research focused on "the change (before- and after-) in the total score on the AMSTAR" and "the change (before- and after-) in each item on it", information in the Results section was presented in a similar format. We hope that this approach is acceptable.

Point 12: In the Discussion of results more previous works should be cited to discuss results.

Response 12: We cited the following four articles in the Discussion:

[18] Loria, A.; Arroyo, P. Language and country preponderance trends in MEDLINE and its causes. J Med Libr Assoc 2005,93,381-5.

[19] Xu, Q.; Boggio, A. Countries’ biomedical publications and attraction scores: a PubMed-based assessment. F1000Res 2015,3,292.doi:10.12688/f100research.5775.2.

[20] Müller, K.F.; Briel, M.; D'Amario, A.; Kleijnen, J.; Marusic, A.; Wager, E.; et al. Defining publication bias: protocol for a systematic review of highly cited articles and proposal for a new framework. Syst Rev 2013,2,34 doi:10.1186/2046-4053-2-34

[21] Chan, A.W.; Tetzlaff, J.M.; Altman, D.G.; Laupacis, A.; Gøtzsche, P.C.; Krleža‒Jerić, K.; et al. SPIRIT 2013 Statement: Defining standard protocol items for clinical trials. Ann Intern Med 2013,158,200‒7.

Point 13: Table 1 should be reorganized.

Response 13: It is an original style by the CAA, so please leave it as is.

Point 14: The references should be formatted following MDPI format.

Response 14: We corrected references to comply with the MDPI’s bibliography style.

We really appreciate all the constructive comments and suggestions to make our paper a more contemporary and reliable published article.

Round 2

Reviewer 1 Report

The additional text throughout the document lends for better comprehension of the study.  I had concerns with Tables 2 and 3 in particular.

As far as Table 3 is concerned, authors typically indicate significant findings (marked with a symbol, i.e. "p<.001^", with the explanation in in the key: "^indicates a significant result") so that the reader does not have to make the interpretation him/herself. For the Supplementary data table, shouldn't column 3a be answered like the others (Y, N, C, N/A) rather than identifying the databases by name? Perhaps the database name could be added to column 3b. Database names could be represented by an alphabetic symbol for ease of reporting, i.e. PMC for PubMed Central. Shouldn't there be a scoring key as well (Y=yes, etc.)?

Reviewer 2 Report

The authors have improved the manuscript that it is now suitable for publication